# Isolation of the Antifungal Compound Alliodorin from the Heartwood of *Cordia elaeagnoides* A. DC. and the In Silico Analysis of the Laccase

**DOI:** 10.3390/plants13101294

**Published:** 2024-05-08

**Authors:** Santiago José Guevara-Martínez, Francisco Villanueva-Mejía, Adalberto Zamudio-Ojeda, Rafael Herrera-Bucio, Fredy Geovannini Morales-Palacios

**Affiliations:** 1Department of Pharmacology, School of Exact Sciences and Engineering, University of Guadalajara, Boulevard Gral. Marcelino García Barragán 1421, Olímpica, Guadalajara 44840, Jalisco, Mexico; santiago.guevara@academicos.udg.mx; 2Instituto Tecnológico de Pabellón de Arteaga, Carretera a la estación de Rincón de Romos, km 1, Aguacalientes 20670, Aguascalientes, Mexico; fvillamejia@gmail.com; 3Deparment of Physics, School of Exact Sciences and Engineering, University of Guadalajara, Boulevard Gral. Marcelino García Barragán 1421, Olímpica, Guadalajara 44840, Jalisco, Mexico; adalberto.zojeda@academicos.udg.mx; 4Instituto de Investigaciones Químico Biólogicas, Universidad Michoacana de San Nicolás de Hidalgo, Francisco J. Múgica, s/n, Morelia 58030, Michoacán, Mexico

**Keywords:** antifungal, extractives, molecular docking

## Abstract

*Cordia elaeagnoides* A. DC. is an endemic species of Mexico valued for its timber. Renowned for its durability, resistance, and versatile applications in medicine, this tree holds significant commercial importance. Tetrahydrofuran (THF) extract from the heartwood of *C. elaeagnoides* was studied. Through chromatographic column purification, the compound 8-(2,5-Dihydroxyphenyl)-2,6-dimethylocta-2,6-dienal, also known as alliodorin, was successfully isolated. Identification of alliodorin was confirmed through comprehensive analysis utilizing NMR, IR, and mass spectrometry techniques. Inhibition tests were conducted using both the THF extract and alliodorin against the rotting fungus *Trametes versicolor* (L.) Lloyd, employing the agar well diffusion assay. Remarkably, alliodorin exhibited 100% inhibition with a median lethal concentration of 0.079 mg/mL and a total lethal concentration of 0.127 mg/mL, in comparison to the commercial fungicide benomyl, which requires a concentration of 1 mg/mL. In silico analysis through molecular docking on the laccase enzyme was proposed in order to explain the inhibitory activity against the fungus *T. versicolor*, as this enzyme is one of the main sources of nutrients and development for the fungus. Based on these findings, we deduced that alliodorin holds promise as a potent antifungal agent, potentially applicable in a wide array of technological and environmentally friendly initiatives.

## 1. Introduction

*Cordia elaeagnoides*, an indigenous species of Mexico, is a member of the Boraginaceae family and is commonly referred to as cueramo, bocote, and bojote. Its natural habitat spans the Pacific slope, encompassing regions from Southern Nayarit to Jalisco, Guerrero, Oaxaca, Chiapas, and extending to the Balsas River basin and Quintana Roo. Typically found in semi-deciduous and low deciduous forests, this species has long been recognized for its ecological and economic significance [1,2]. With its wide distribution and cultural importance, *C. elaeagnoides* plays a vital role in the biodiversity and traditional practices of the regions it inhabits.

Renowned for its dense, heavy wood characterized by its captivating color and grain, *C. elaeagnoides* holds immense value for various applications. Its sturdy timber is coveted for crafting mosaic-styled parquet floors, ornamental veneers, high-end furniture, cabinetry, and a diverse range of artisanal goods including salt shakers, snack boxes, spice racks, chess pieces, and intricately carved utensil handles [2]. Recent studies have highlighted the exceptional durability of *C. elaeagnoides* wood, further enhancing its desirability for both functional and decorative purposes.

Durability tests conducted across three distinct natural environments have consistently classified it as ‘very durable’ according to the European Standard EN 350-1 [3,4]. Moreover, its heartwood has demonstrated resistance ranging from moderate to high against decay fungi, earning classifications of Class 2 and 3 according to ASTM D 2017-81 standards as per CONAFOR. These assessments underscore the robustness and resilience of the material under various environmental conditions, making it a promising choice for applications requiring long-term performance and resistance to fungal decay.

The term ‘Durability’ in wood refers to its inherent ability to withstand attacks from various sources, including xylophagous fungi, bacteria, insects, marine borers, and chemical and mechanical wear [5,6]. This resilience is largely attributed to chemical components known as extractives or secondary metabolites, encompassing a range of substances such as terpenes, phenols, tannins, aliphatic and aromatic hydrocarbons, essential oils, fatty acids, resins, and minerals [7,8].

Wood comprises cellulose and hemicellulose fibers forming the framework, along with lignin providing rigidity. Additionally, substances produced during sapwood maturation, the active physiological part of the trunk, contribute to wood composition. As the sapwood matures into heartwood, it serves as structural support for the tree while storing compounds (extractives) that impart characteristics like color, odor, and resistance to degradation and pathogen attacks. These extractives can be extracted using organic solvents, hence the name ‘extractives.’ Understanding the role of extractives in wood durability is paramount for industries reliant on wood products for their performance and longevity [7,9,10,11].

Among the various agents of wood biodeterioration, xylophagous fungi, specifically Basidiomycetes responsible for white and brown rots, pose the most frequent and aggressive threat to wood in use. These fungi possess the ability to degrade the primary components of wood through enzymatic biological processes [12,13]. Consequently, the durability of wood is often correlated with its resistance to fungal degradation. Understanding the mechanisms by which these fungi attack wood and identifying strategies to mitigate their impact are crucial for preserving the structural integrity and longevity of wood-based materials in various applications [14].

The three types of fungal decay are soft rot, brown rot, and white rot. Soft rot, although less common, is characterized by the degradation of the cellulose of the cell wall, causing softening of the wood, but it requires high levels of moisture, so it is not very common [15]; brown rot is characterized by the depolymerization of cellulose and hemicellulose, resulting in the wood turning a reddish-brown color, without altering the lignin and is commonly observed in various timber and forests ecosystems [13,16,17,18]; and white rot primarily decomposes lignin, but some fungi of this type of decay have the ability to decompose cellulose and hemicellulose simultaneously with lignin [19,20,21].

Notably, fungus *Trametes versicolor* (L.) Lloyd, known as turkey tail due to its concentric multicolored morphology, found in all temperate zones of Asia, America, and Europe including the United Kingdom, is the most common and aggressive white rot fungus of wood, due to its capacity to break down and metabolize lignin and wood carbohydrates [22,23,24,25]. Consequently, American standards ASTM D 2017-81 [26] advocate testing against *T. versicolor* in laboratory durability experiments due to these factors. This type of degradation is mainly due to the extracellular enzyme produced by the fungus called laccase, which contains four copper (Cu) atoms classified as types 1, 2, and 3. These atoms initiate an oxidation–reduction process, starting with type 1 copper (T1), which oxidizes organic substrates (*ortho*-, *para*-, and diphenols, aminophenols, lignin dimers, etc.) and inorganic substates (cypermethrin, carbofuran nematicide, polychlorinated biphenyls, pesticides, pentachlorophenols, etc.) [27,28], followed by type 2 and two type 3 copper atoms that form a trinuclear copper cluster (TNC). In this TNC, oxygen obtained from the oxidation of T1 is reduced to water. Through this enzyme process, the fungus obtains nutrients for its development [29,30,31,32,33]. Therefore, some studies suggest that laccase inhibitors are small molecules capable of binding to the Cu ions of the TCN, thereby interrupting the internal oxidation process such as sodium azide, some small halides, heavy metals, and ethylenediaminetetraacetic acid (EDTA) [34,35].

Fungal degradation can be assessed through laboratory methods or accelerated tests, which are known for their shorter duration. Among these methods are the plate diffusion and agar well diffusion assay techniques, where a xylolytic strain is cultured in an artificial medium (agar) dispensed in Petri dishes. Compounds (extracts) are added at varying concentrations, and the growth inhibition zone is measured relative to a control [36,37,38]. These assays provide valuable insights into the effectiveness of the tested compounds against fungal degradation and offer a convenient means of screening potential antifungal agents. By rapidly assessing the inhibitory effects of different compounds on fungal growth, researchers and industries can identify promising candidates for further investigation or application in wood preservation and protection strategies.

## 2. Results

### 2.1. Extractives Content

The quantity of extractables obtained from heartwood flour via Soxhlet extraction was determined to be 18.7%. Out of this total quantity of extractables, 1.84% was extracted using hexane, an apolar solvent, while 16.87% was extracted using THF, a medium polarity solvent. Given the absence of reported data regarding the total extractable content from *C. elaeagnoides* trees, this yield is considered satisfactory, especially considering that the literature suggests an average extractable content ranging from 10% to 20% in wood [39].

### 2.2. Antifungal Activity

The inhibition test was conducted using the plate diffusion method with the THF and hexane extracts. In this method, potato dextrose agar was used, and the entire Petri dish was utilized. After sterilizing the medium, extracts were incorporated at concentrations of 250 and 500 mg/L, respectively. Once the medium solidified, it was inoculated with the mycelium of the fungus *T. versicolor* and incubated for 7 days at 28 ± 2 °C. Similarly, a control consisting of untreated agar was prepared and one more with methanol was used as the negative control. Subsequently, the percentage of inhibition was determined using the agar well diffusion assay formula methodology described earlier. The concentrations used were determined based on previous studies [40,41].

The results revealed that the THF extract exhibited complete inhibition at a concentration of 500 mg/L and 66% inhibition at 250 mg/L, whereas the hexane extract displayed 60% inhibition at 500 mg/L. Consequently, the THF extract was selected for further purification via column chromatography to isolate the active component(s) responsible for this inhibitory activity. This rigorous process aims to identify and characterize the specific compounds within the extract that contribute to its antifungal properties, potentially leading to the development of novel bioactive agents for wood preservation or other applications.

Subsequently, the agar well diffusion assay method was studied because it was more practical, cost-effective, and optimal for obtaining the median lethal concentrations (LC50s) and the total lethal concentrations (LC100s) [42]. Additionally, it allowed comparisons with a commercial fungicide benomyl [methyl 1-(butylcarbamoyl) benzimidazole-2-yl-carbamate] (Figure 1). This compound is used in agriculture and horticulture to eliminate entomopathogenic infections, mainly fungi. It is worth mentioning that this compound has been discontinued in many countries in recent years due to reports of diseases and health damage [43]. The agar well diffusion assay presents a valuable alternative for assessing the efficacy of natural compounds against fungal pathogens, especially in light of concerns regarding the safety and environmental impact of synthetic pesticides like benomyl.

Subsequent purification yielded a dark brown-colored oil, which was subjected to inhibition tests in triplicate at concentrations of 0, 0.03, 0.06, 0.075, 0.1, and 0.125 mg/mL. Notably, the complete inhibition of the fungus was observed at a concentration of 0.125 mg/mL (Figure 1).

### 2.3. Identification of the Crystalline Isolated Compound

The THF extract was subjected to purification on a 1 g chromatographic column, using silica gel as the stationary phase and a mobile phase consisting of hexane/EtOAc mixtures in ascending polarity, employing 150 mL per solvent mixture. In fractions with ratios ranging from 9:1 to 7:3 hexane/EtOAc, a major compound was observed via thin-layer chromatography (TLC), obtained through isolation using a 9:1 hexane/EtOAc polarity, and was subjected to characterization via NMR spectroscopy, yielding the following results (Figure 2 and Figure 3):

^1^H NMR (400 MHz, CDCl_3_) δ 1.74 (s, 3H, H-10′), 1.75 (s, 3H, H-9′), 2.26 (t, J = 7.2 Hz, 2H, H-4′), 2.51 (q, J = 7.3 Hz, 2H, H-5′), 3.30 (d, J = 7.2 Hz, 2H, H-1′), 4.87 (s, 1H, OH), 5.14 (s, 1H, OH), 5.32 (tq, J = 8.5, 1.2 Hz, 1H, H-2′), 6.49 (tq, J = 7.3, 1.3 Hz, 1H, H-6′), 6.54 (d, J = 3.0, Hz, 1H, H-6), 6.57 (dd, J = 8.4, 3.0 Hz, 1H, H-4), 6.65 (d, J = 8.4 Hz, 1H, H-3), 9.36 (s, 1H, H-8′). ^13^C NMR (101 MHz, CDCl_3_) δ 9.28 (C-10′), 15.98 (C-9′), 27.09 (C-5′), 28.95 (C-1′), 37.91 (C-4′), 113.72 (C-6), 116.22 (C-3), 116.35 (C-4), 123.10 (C-2′), 128.05 (C-3′), 135.82 (C-1), 139.58 (C-7′), 147.56 (C-5), 149.47 (C-2), 154.70 (C-6′), 196.09 (C-8′), Rf: 0.22 (Hex: EtcOAc, 8:2).

Furthermore, electron impact mass spectrometry showed a molecular ion of 259.3834 *m*/*z*, corresponding to the calculated [M-1] 260.313 of the formula C_16_H_20_O_3_. Additionally, the infrared analysis showed IR vmax/cm^−1^: 3352, 2859, 2973, 1664. The data obtained from ^13^C NMR were analyzed and compared with the reported data obtained from *Cordia polycephala* [44,45] (Table 1), which corresponded to 8-(2,5-Dihydroxyphenyl)-2,6-dimethylocta-2,6-dienal or alliodorin (Figure 4).

### 2.4. Analysis of the Docking Study and the Laccase–Alliodorin Complex

The docking study and the laccase–alliodorin complex yielded significant insights. Alliodorin demonstrated inhibitory activity against the fungus *T. versicolor*, akin to the THF extract. While numerous studies have reported the inhibitory activity of different extracts from various trees, such as hexane and methanolic extracts from the woods *Dalbergia granadillo* Pitt. and *Enterolobium cyclocarpum*, aqueous extracts from the woods of *Mimosa biuncifera* and *Acacia angustissima*, and bark extracts from *Sorbus commixta* [46,47,48,49], only a select few have isolated components capable of such activity. For instance, medicarpin isolated from *Dalbergia congestiflora* Pittier exhibited inhibitory activity against *T. versicolor* at a concentration of 150 mg/L [41].

A molecular docking analysis of the mechanism of action of medicarpin on the laccase enzyme elucidated interaction and inhibition by obstructing the oxygen entry site towards the TNC and the T1 cavity of the laccase enzyme [50]. Additionally, in in silico studies, enzyme inactivation is attributed to the interaction of compounds with the amino acids covering the copper atoms of T1 and TNC, thereby reducing their potential for oxidation and reduction. Specifically, the amino acids primarily covering Cu T1 include Cys453, Ile455, His458, His395, and Phe463, while TNC is covered by His64, His66, His111, His109, His398, His400, His452, and His454 [51,52,53]. Given these references, blind docking was performed with alliodorin on laccase to assess the probability of interaction at reported sites and amino acids.

## 3. Discussion

The findings regarding the extractive contents strongly support the decision to utilize THF (tetrahydrofuran) as the solvent for extraction. Our research group observed a significantly higher yield of extractables with THF compared to hexane and ethyl acetate, highlighting its effectiveness, and we optimized the extraction protocols and the yield of bioactive compounds for further analysis and potential application in various fields such as pharmaceuticals, agriculture, and wood preservation.

Furthermore, Soxhlet apparatus was chosen for the isolation of wood extractives due to its superior yields compared to other methods such as maceration processes, heat-assisted maceration, and reflux with different solvents. Additionally, it is recommended by the TAPPI T-264 cm-97 standard for the isolation of extractives [54]. The Soxhlet extraction method offers several advantages, including efficient extraction of target compounds, reproducibility, and ease of operation. Its compatibility with a wide range of solvents makes it suitable for extracting various types of compounds from diverse sample matrices.

Regarding the modification of antifungal methods employed, while the plate diffusion method remains specific and approved by ASTM standards for conducting durability tests (fungi) on wood, integrating new methodologies for fungal inhibition is crucial. This integration generates more viable options in terms of measurement, cost, ease of use, and standardization of methodology. An example is the agar well diffusion assay method, which is a modification of the disc diffusion method using microdilutions. The agar well diffusion assay offers several advantages over traditional methods, including the ability to assess multiple samples simultaneously, improved accuracy in measuring inhibition zones, and enhanced sensitivity to detect antifungal activity.

The use of microdilutions has been approved by the Antifungal Susceptibility Subcommittee of the Clinical and Laboratory Standards Institute (CLSI; formerly NCCLS), enabling the use and standardization of various concentrations of the fungus. Moreover, it is a reliable, simple measurement method that demonstrates the susceptibility and reproducibility of the extracts for the fungus at different concentrations in microdilutions compared to alternative methods [55,56]. The endorsement of microdilution techniques by authoritative bodies like the CLSI underscores their credibility and utility in assessing antifungal activity.

Based on the data from the inhibitions, a median lethal concentration curve (LC50) of 0.079 mg/mL and LC100 were generated, as depicted in Figure 2. Comparatively, when contrasted with the commercial fungicide benomyl, which necessitates 1 mg/mL to achieve 100% inhibition, the activity of the alliodorin compound emerges as highly effective.

Furthermore, thin-layer chromatography (TLC) tests unveiled a notable distinction between the hexane and THF extracts: the absence of dark brown oil substance in the hexane extract. This observation hints at the presence of other inhibitory components in the hexane extract, which could potentially contribute to the inhibition results obtained with this extract. This indicates the complexity of the inhibitory mechanisms at play and underscores the need for further investigation into the composition and activities of the various components within the extracts.

The in silico study using molecular docking was carried out through blind docking, which involved allowing the ligand (alliodorin) to interact freely and randomly without restrictions on the entire protein (laccase: 1GYC). This was performed to determine if the ligand interacts probably on the sites already reported for its inactivation. The results obtained from AutoDock mostly demonstrated three anchoring sites of alliodorin on laccase, which are the T1 site, the oxygen entrance channel, and a site near the TNC (Figure 5). These sites are reported in the literature as sites of inactivation for this protein. It is worth mentioning that the binding energy at the T1 site was −6.19 kcal/mol, which is lower than that reported for medicarpin, as its binding energy at this site was −7.35 kcal/mol. However, at the TNC site, the best binding energy was observed, being −7.66 kcal/mol. No anchoring is reported for medicarpin at this site. Regarding the oxygen entrance site, the binding energy of alliodorin was −7.31 kcal/mol, which was higher than that reported for medicarpin (−7.24 kcal/mol). Since the anchoring has been demonstrated at the sites of interest of alliodorin–laccase, the interactions observed at these sites are described below.

Molecular docking results revealed anchoring in the T1 cavity, with notable proximity to Cu and the formation of a hydrogen bond with His458, along with interaction through Van der Waals forces with Ile455, both of which cover T1 (Figure 6). Additionally, of significant interest was the anchoring of alliodorin in a site very close to TNC, which resulted in Van der Waals interactions with two of the coppers, as well as with the amino acids covering them, such as His111, His109, and His452. Moreover, strong interactions (hydrogen bonds) were observed with Ser113, Arg157, and Glu460 (Figure 7). Based on these findings and reported information, it is evident that alliodorin interacts at the sites of interest or causes the inactivation of the laccase enzyme with better interactions than those reported for medicarpin [50]. This suggests that the inhibition mechanism of alliodorin involves obstructing the T1 and TNC through interaction with the amino acids near the copper atoms, akin to medicarpin but with greater efficiency. This conclusion is supported due to the strong interactions and hydrogen bond formations observed in the complexes formed between alliodorin and the laccase enzyme. Overall, these results provide insights into the molecular mechanisms underlying the inhibitory activity of alliodorin against the laccase enzyme.

## 4. Materials and Methods

### 4.1. Obtaining Wood Extracts

*C. elaeagnoides* was collected in Taretan, Michoacán, Mexico, in its natural habitat. The tree was identified by the Faculty of Wood Technology Engineering at the Universidad Michoacán de San Nicolás de Hidalgo in Morelia, Michoacán. The heartwood of the *C. elaeagnoides* tree was air-dried, and it was reduced in size with a chisel and hammer into splinters to be fed into a mill (IPHARMACHINE YF3-1 brand with a capacity of 5 Kg/h) and ground into flour. Subsequently, the batches of 16 g were subjected to extractions using a Soxhlet apparatus, refluxing for 4 h with hexane and THF, in 250 mL of each solvent [57]. The THF extract was purified using column chromatography with hexane/EtOAc 90:10 mixtures, isolating the compound alliodorin, which was identified by ^1^H and ^13^C NMR, IR, and mass spectrometry.

The nuclear magnetic resonance spectra of ^1^H (^1^H-NMR) and ^13^C (^13^C-NMR) were obtained at the Institute of Chemical-Biological Research using a Varian Mercuri Plus spectrometer at 400 MHz and a Varian Mercuri Plus spectrometer at 200 MHz. Infrared spectra (IR) were determined using a Thermo Scientific Nicolet iS10 spectrophotometer employing the ATR technique. The mass spectrum was obtained using a Thermo Scientific ISQ CT spectrometric equipment employing electron ionization impact.

### 4.2. Antifungal Bioassay

The determination of antifungal activity of alliodorin was carried out using the agar well diffusion assay method [55,58,59], which involves creating 5 mm perforations in 3.9% potato dextrose agar. In each well, 50 µL of alliodorin at concentrations of 0, 0.03, 0.06, 0.075, 0.1, and 0.125 mg/mL was added, along with the positive control, the fungicide benomyl at 1 mg/mL [60,61], and the negative control, methanol, which was used as a vehicle. Subsequently, a fragment of *T. versicolor* fungus (voucher number: ATCC-32745) was inoculated into each well and incubated for 8 days at 28 ± 2 °C. The maximum growth was observed in the negative control or 0% inhibition was considered as the control for comparison (Figure 1). Each treatment was replicated three times. The percentage of inhibition was calculated using the formula outlined by Rutiaga [40]:(1)$$\%\;\mathrm{inhibition}=\left(\frac{\mathrm{growth}\;\mathrm{control}-\mathrm{growth}\;\mathrm{treatment}}{\mathrm{growth}\;\mathrm{control}}\right)\times 100$$

This formula compares the average diameter of fungal growth in the treatment wells to that of the negative control well and expresses the inhibition percentage accordingly.

Using the results of inhibition percentages, a concentration–response curve was constructed to obtain the median lethal concentrations (LC50s) and the total lethal concentrations (LC100s) of alliodorin.

### 4.3. Modeling and Molecular Docking

The construction of the alliodorin compound was conducted using Gaussian 16 was employed for a conformational analysis using density functional theory (DFT) method with the B3LYP hybrid functional and a 6-311G** basis set [62].

The three-dimensional structure of the laccase enzyme from the fungus *T. versicolor* was obtained from the Protein Data Bank (PDB), with the code 1GYC [63]. Molecular docking was performed using the open-source AutoDock 4.2 software [64]. Initially, blind docking was conducted using AutodockTools. Polar hydrogen atoms were added, Kollman partial charges were assigned, and the Lamarckian genetic algorithm (LGA) was employed for the conformational search of the protein–ligand, with a random population of substrate conformations of up to 250 arbitrary orientations, a mutation rate of 0.02, and a crossover rate of 0.8, and simulations were conducted considering 2.5 million energy evaluations with a maximum of 27,000 generations. Each simulation was performed 1000 times, generating 1000 poses to obtain the binding energy of the complex and determine if alliodorin interacts in the reported sites of interest in a free search [65]. Additionally, the process automatically assigns the number of allowed torsions for alliodorin, favoring more flexible poses in molecular docking. Interaction specifications on the protein 1GYC were considered by centering a cubic box composed of 126 × 126 × 126 grid points with a spacing of 0.375 Å between each point. It is worth mentioning that the force field implemented by AutoDock does not include parameters for Cu atoms or their ionic forms, so the AD4_parameters.dat file was modified to include Cu parameters according to the instructions in the AutoDock 4.2 user guide. And with BIOVA Discovery Studio Visualizer, the interactions with amino acids at the binding sites were visualized.

## 5. Conclusions

The findings of this study suggest that the natural durability of *C. elaeagnoides* heartwood cannot be solely attributed to its hardness and density, but also to the content of its extractables. This is evidenced by the inhibitory activity displayed by the hexane and THF extracts, as well as the isolated compound alliodorin, against one of the most aggressive fungi known for wood degradation.

Moreover, simulation studies play a crucial role in elucidating interaction mechanisms and identifying binding sites, as demonstrated with alliodorin. This analysis revealed that alliodorin inhibits the laccase enzyme by anchoring near the copper atoms of T1 and TNC, thus impeding substrate degradation and resource formation necessary for the development of the fungus *T. versicolor*.

In conclusion, alliodorin exhibits inhibitory activity against white rot fungus, shedding light on its potential role in enhancing the natural durability of *C. elaeagnoides* heartwood. With the results obtained in this study, we believe that research and development of preservative and/or antifungal formulations based on natural products can be pursued.

## Data Availability

Data is contained within the article.

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
