# Peer review of "Isolation of the Antifungal Compound Alliodorin from the Heartwood of Cordia elaeagnoides A. DC. and the In Silico Analysis of the Laccase"

_plants, 2024, doi:10.3390/plants13101294_

Round 1
Reviewer 1 Report
Comments and Suggestions for Authors
This paper describes an intersting antifungal activity of alloidorin and investigates a possible machanism of action on laccase by docking studies.
In my opinion, the overall scientific quality can be considered acceptable if a few major modifications will be considered:
1. The traditional nema of the main compound investigated is Alliodorin, not Aliodorin.
2. Hexane is NOT a medium polar solvent. It is definitely a apolar solvent! (page 3): please rewrite tihis sentence.
3. Section 2.2: concentration is really 500 mg/mL or perhaps 500 mg/L?
4. Same section: which is the solvent for antifungal test?
5. Same section, page 4: “honey-like” is not an acceptable descriptor for a chemical compound. Cite directly the chemical identity.
6. The heading of section 2.4 and the first sentence: “the analysis of docking study…” is reduntant. Write directly “docking study”.
In my opinion, a minor revision is also necessary for the “Discussion” section. It is sometime exceedingly verbose:
Lines 234-243: THF IS chemically an ether. It is not a mere matter of classification! Further, the entire paragraph lack of any significance as the choice of a solvent is a trivial fact in the natural product laboratory. Please, reduce at a maximum of 3, 4 lines.
Lines 244 – 246: this assertion is not well grounded. The better technique depends upon the goal. I would suggest to remove it.
Line 264: please, clarify in which sense is comparable.
Lines 268 – 272: What can one “understand” by the linear correlation reported in graph 2? Also this part is verbose. The result obtained is just Alliodorin being XXX times more potent than benomyl. That is all we can argue from graph 2.
Line 276: again “honey-like substance….”
Line 303: add a reference, please.
Lines 307 – 310: the same thing is repeated twice.
Lines 315 – 317: a specimen voucher deposit is necessary.
Line 321: specify the solvent volume.
References section: check carefully the completeness and the uniformity of books references.
Author Response
Reviewer #1:
Query 1. The traditional name of the main compound investigated is Alliodorin, not Aliodorin.
Response 1. We sincerely appreciate the review time dedicated by Reviewer #1 to our manuscript, her/his insightful comments and suggestions significantly enhance the scientific precision of our specimen description, which was corrected and incorporated as advised.
Query 2. Hexane is NOT a medium polar solvent. It is definitely a apolar solvent! (page 3): please rewrite this sentence.
Response 2. We thank the reviewer for the assistance in this comment. Appropriate modifications in the text are included in line 122-123.
Query 3. Section 2.2: concentration is really 500 mg/mL or perhaps 500 mg/L?
Response 3. We thank the reviewer for this comment. This improved version of our manuscript provides data specification as 500 mg/L.
Query 4. Same section: which is the solvent for antifungal test?
Response 4. We thank the reviewer for this comment and have followed the suggestion in line 133.
Query 5. Same section, page 4: “honey-like” is not an acceptable descriptor for a chemical compound. Cite directly the chemical identity.
Response 5. We appreciate the reviewer’s comment, and we have addressed the suggestion by specifying the chemical identity of the oil.
Query 6. The heading of section 2.4 and the first sentence: “the analysis of docking study…” is reduntant. Write directly “docking study”.
Response 6. We thank the reviewer for pointing out this redundancy. We have revised the text accordingly, changing it in line 183 as per your suggestion.
Query 7. Lines 234-243: THF IS chemically an ether. It is not a mere matter of classification! Further, the entire paragraph lack of any significance as the choice of a solvent is a trivial fact in the natural product laboratory. Please, reduce at a maximum of 3, 4 lines.
Response 7. We thank you for the careful and insightful review. Specifically, we want to emphasize that the utilization of THF is highly pertinent to the journal's aims and scope. This is because it represents a novel approach not previously documented in the literature for the extraction of extractables/secondary metabolites. Additionally, our methodology was further enhance by the modification of dichloromethane and ethyl acetate solvents, with the latter proving superior in yield compared to the combined yields than the two solvents mentioned previously.
Query 8. Lines 244 – 246: this assertion is not well grounded. The better technique depends upon the goal. I would suggest to remove it.
Response 8. We thank the reviewer for her/his thoughtful comments and efforts toward improving our manuscript. In response, we have streamlined the verbose sections and specified the necessity of this statement for woods, supported by literature reports demonstrating its superior yield and ease of use. The changes can be observed in the lines 240-242.
Query 9. Line 264: please, clarify in which sense is comparable.
Response 9. We are grateful to the reviewer for their thoughtful comments and dedication to enhancing our manuscript. In response, we have duly corrected and incorporated as advised at line 259-261.
Query 10. Lines 268 – 272: What can one “understand” by the linear correlation reported in graph 2? Also this part is verbose. The result obtained is just Alliodorin being XXX times more potent than benomyl. That is all we can argue from graph 2.
Response 10. We value the reviewer’s feedback. We have revised this paragraph to improve clarity and conciseness. As your suggested, we now focus on the main finding from graph 2, which is the focus between the median lethal concentrations (LC50) and the total lethal concentrations (LC100) on both compounds. The changes can be observed in the lines 265-268.
Query 11. Line 276: again “honey-like substance….”
Response 11. We appreciate the reviewer’s comment, and we have addressed the suggestion by specifying the chemical identity of the oil, line 270.
Query 12. Line 303: add a reference, please.
Response 12. We value the reviewer’s feedback. It was added in line 299.
(50) Martínez-Sotres C, Rutiaga-Quiñones JG, Herrera-Bucio R, Gallo M, López-Albarrán P. Molecular docking insights into the inhibition of laccase activity by medicarpin. Wood Sci Technol. 2015 Jul 8;49(4):857–68. https://doi.org/10.1007/s00226-015-0734-8
Query 13. Lines 307 – 310: the same thing is repeated twice.
Response 13. We thank the reviewer for pointing out this redundancy. We have revised the text accordingly, changing it in line 265-268 as per your suggestion.
Query 14. Lines 315 – 317: a specimen voucher deposit is necessary.
Response 14. We deeply appreciate the reviewer’s meticulous and insightful evaluation. We have diligently addressed all the revision concerns, including the acknowledgment that tree voucher was provided and identified by the Wood Anatomy Laboratory belonging to the Faculty of Wood Technology Engineering at the Universidad Michoacana de San Nicolás de Hidalgo. This identification was conducted by Professor Ing. Teresa García Moreno, as seasoned expert with over 20 years of experience in xylotomy (wood identification and research).
Query 15. Line 321: specify the solvent volume.
Response 15. We are grateful for the reviewer’s meticulous evaluation of the manuscript. We have included the solvent specification on lines 313-314.
Query 16. References section: check carefully the completeness and the uniformity of books references.
Response 16. I appreciate your suggestion, as a result of which additional data has been added for the uniformity and identification of the mentioned books, following the author guidelines provided by the journal.
Reviewer 2 Report
Comments and Suggestions for Authors
The manuscript "Isolation of the antifungal compound Aliodorin from the heart wood of Cordia elaeagnoides and the in-silico analysis of the laccase" is dealing with identifications of compounds from Cordia elaeagnoides extract, with special attention to aliodorin, important antifungal compound. Here are some comments for improving the manuscript:
Cordia elaeagnoides in italic
When you first time mention plant species, please provide full name: Cordia elaeagnoides DC., after that use common abreviation C. elaeagnoides.
The name of fungus Trametes Versicolor correct to Trametes versicolor (L.) Lloyd, in italic! Same as previous, the abbreviation is T. versicolor (for second and other time mention in the manuscript).
Please provide voucher species for C. elaeagnoides, and fungal strain number for T. versicolor in material and method part.
Author Response
Reviewer #2:
Query 1. Cordia elaeagnoides in italic
Response 1. We sincerely appreciate the review time dedicated by Reviewer #2 to our manuscript, her/his insightful comments and suggestions significantly enhance the scientific precision of our specimen description, which was corrected and incorporated as advised.
Query 2. When you first time mention plant species, please provide full name: Cordia elaeagnoides DC., after that use common abreviation C. elaeagnoides.
Response 2. We thank the reviewer for the assistance in this comment. We have incorporated appropriate modification throughout the manuscript as per the provided comment.
Query 3. The name of fungus Trametes Versicolor correct to Trametes versicolor (L.) Lloyd, in italic! Same as previous, the abbreviation is T. versicolor (for second and other time mention in the manuscript).
Response 3. We deeply appreciate the reviewer’s meticulous and insightful evaluation. We have diligently addressed all the revision concerns. Specifically, we added the voucher number for the fungus in line 330. Additionally, we included the acknowledgment that tree voucher was provided and identified by the Wood Anatomy Laboratory belonging to the Faculty of Wood Technology Engineering at the Universidad Michoacana de San Nicolás de Hidalgo. This identification was conducted by Professor Ing. Teresa García Moreno, as seasoned expert with over 20 years of experience in xylotomy (wood identification and research).
Reviewer 3 Report
Comments and Suggestions for Authors
The themes of the publication submitted for review, entitled: "Isolation of the antifungal compound Aliodorin from the heartwood of Cordia elaeagnoides and the in-silico analysis of the laccase" concerns the isolation of a compound called aliodorin. This compound has antifungal activity against T. versicolor. The authors also showed that aliodorin inhibits the laccase enzyme. The publication is interesting and shows that plant raw materials may be important in the search for biologically active compounds.
However, I have a few minor comments:
1) In the line 123 "These solvents are classified as having medium polarity." I think only THF, but not hexane. Please correct it.
2) In the line 168 there is 1H RMN, it should be 1H NMR.
3) In the lines 187 and 190, the names of tree should be written in italics (and also in the lines 38, 44, 47, 51, 91, 315, 372).
4) In the line 323, 1H and 13C are written as normal index, not superscript.
5) In the name of Figure 2 there is: NMR-H1, it should be 1H NMR.
6) I think the 1H NMR spectrum could contain the integrationof hydrogen atoms of individual signals and it would be good if it were larger.
Author Response
Reviewer #3:
Query 1. In the line 123 "These solvents are classified as having medium polarity." I think only THF, but not hexane. Please correct it.
Response 1. We sincerely appreciate the review time dedicated by Reviewer #3 to our manuscript, her/his insightful comments and suggestions significantly enhance the scientific precision, appropriate modifications in the text are included in lines 122-123 as advised.
Query 2. In the line 168 there is 1H RMN, it should be 1H NMR.
Response 2. We thank the reviewer for the assistance in this comment. We have incorporated appropriate modification in the line 168 and throughout the manuscript as per the provided comment.
Query 3. In the lines 187 and 190, the names of tree should be written in italics (and also in the lines 38, 44, 47, 51, 91, 315, 372
Response 3. We deeply appreciate the reviewer’s meticulous and insightful evaluation. We have carefully addressed all the revision concerns raised throughout the manuscript in accordance with the provided comments.
Query 4. In the line 323, 1H and 13C are written as normal index, not superscript.
Response 4. We thank the reviewer for her/his thoughtful comments and efforts toward improving our manuscript. In response, we have made the necessary changes as suggested.
Query 5. In the name of Figure 2 there is: NMR-H1, it should be 1H NMR.
Response 5. We appreciate the reviewer’s comment, and we have addressed the suggestion by specifying as 1H NMR.
Query 6. I think the 1H NMR spectrum could contain the integration of hydrogen atoms of individual signals and it would be good if it were larger
Response 6. We appreciate the reviewer’s thoughtful comments and efforts to improve our manuscript. In the 1H NMR spectrum, the display range was expanded, and integrals were included.